# Head Pose as a Proxy for Gaze in Virtual Reality

Padraig Higgins
*CSEE*
*Univ. of MD, Baltimore County*
phiggin1@umbc.edu

Ryan Barron
*CSEE*
*Univ. of MD, Baltimore County*
ryanb4@umbc.edu

Cynthia Matuszek
*CSEE*
*Univ. of MD, Baltimore County*
cmat@umbc.edu

*Abstract*—In this work, we posit that a user's head pose can serve as a proxy for gaze in a VR object selection task. We describe a study in which participants were asked to describe a series of objects in a known order, providing approximate labels for the focus of attention. The participants' head pose was then evaluated as a function of the position and orientation of the headset, and how closely that pose matched the location of known objects was calculated. The object that most closely matched the gaze was then evaluated using a mean reciprocal ranking. We demonstrate that using a concept of gaze derived from head pose can be used to effectively narrow the set of objects that are the target of participants' attention.

*Index Terms*—sim2real, gaze tracking, data generation, virtual reality

## I. Introduction

Predicting the target of human attention is a complex problem, incorporating gesture, language, and gaze, among other signals. Gaze tracking, while valuable, generally requires specialized hardware and often negatively affected by external features such as eye color. Since most commodity virtual reality (VR) hardware does not incorporate gaze-tracking capabilities, using gaze to help understand what a user is referring to in a virtual setting has generally not been widely studied. However, embodied agents that interact with people need to understand and display appropriate responses to human interactions, which depend on a variety of modalities, including speech, gesture, and gaze.

One approach to engaging in such interactions is to communicate via natural language; when language refers specifically to the physical environment in which the robot operates, it may be *grounded* in the context provided by sensors and actuators [7]. When grounded language is used to teach robots about objects in the environment, it is necessary to understand the specific target in the environment to which a person is referring, which can be accomplished or assisted, in part, by tracking gaze [24, 17] to identify the objects that are being described.

Robotic learning is often in the form of machine learning, which requires large amounts of data to properly weight neural connections between the machine learning model's layers [25]. Generating test instances in human-robot interactions can take significant time. One way to optimize data collection in robotic learning is to create a simulation of robot operation, increasing the radius of possible participants, minimizing travel, and reducing machine breakdowns. A further optimization is to use attributes of the human-robot interaction to label the data points as they are collected.

In this work, we hypothesize that a VR participant's head pose can serve as a proxy for gaze to label data. These instances can then be used for training a machine learning model, building an unsupervised learning pipeline for object descriptions. We describe the approach to evaluating head pose in simulation and present the results of mean reciprocal ranking of selected objects to determine the effectiveness of this approach.

## II. Related Work

We build directly on previous work on using vision, language, and gesture to select objects in a scene [16], which found that a combined approach was more effective than any single modality. We use the simulator architecture of [10], which incorporates ROS, Unity, the human, and the learning model, as well as the robot and virtual reality technology utilized. Learning models in simulation and then transferring them to a physical system is in the broad area of simulation-to-reality, or sim2real, studies; sim2real approaches have had limited applicability to human-robot interaction (HRI) because of the difficulty of simulating the complexities of human interaction [11]. We address this difficulty by using VR to incorporate people in the simulation environment.

Robots that have a method to express gaze are perceived more favorably and perform better during interactions with humans [18, 22, 1]. It has also been shown to be a useful tool to measure a person's engagement [6] during an interaction and as a measure of the person's perception of the robot [12]. Gaze has been used as a tool to improve robotic manipulation and handoff tasks, where gaze provides insight to the human participant's intent [4, 3]. It has also been used to establish and maintain a common ground during interactions [17, 19].

The performance of eye tracking in VR has been compared to eye tracking in the real world under ideal circumstances [21]. The accuracy did not differ when gazing as static targets, and only showed small differences at targets at varied distances, but did show larger differences when tracking moving target, and showed that the precision in VR was much worse when focusing on static targets. This work only investigated the performance of eye tracking, fixing the head in place in both conditions.

This material is based in part upon work supported by the National Science Foundation under Grant Nos. 1940931 and 2024878.

There has been work that has shown that head pose by itself cannot replace eye gaze, consistent with intuition [20, 13]. However as not all VR headsets support eye tracking, and as gaze tracking in human-robot interactions is not always viable, we hope to compare how well head pose works as a proxy for gaze in VR on the specific task of object selection when a person is teaching a robot about aspects of the environment.

Work has shown that head pose can be used as a method of control for user interfaces in both virtual [9] and augmented [8] reality. The use of bidirectional gaze in interactions with a virtual agent can improve task performance, and it has been shown that using head pose as a proxy for gaze in a system utilizing bidirectional gaze performs similarly to using full eye tracking [2].

## III. METHODS

In this work, gaze via head pose tracking is evaluated as a source of labels for objects, with the goal of allowing the robot to determine on which object a participant's attention is fixed. We briefly describe the simulation environment RIVR [10], then discuss the data collection performed and give details for how the gaze vector is calculated.

### A. Simulator

This experiment was conducted using RIVR [10], a virtual reality robotics simulator. RIVR is constructed out of three components: a VR client, a render server, and a server running ROS. The VR Client provides the virtual reality representation of an environment for the human and robot to interact in. For this experiment a kitchen environment from AI2Thor [14] was used as a base. The virtual reality client is the only part of the system that is run on a participant's local machine. It streams audio captured by the headset, the positions of the headset and controllers along with all the interactive objects in the scene, and button inputs to the ROS node for the robot. Only the participants' view is rendered and streamed to the participant's headset, while the Unity Render Server models the robot's more complex sensors on a remote server.

The render server uses Unity to render the environment and the effects of actions. This server takes as input the robot state from the ROS node, as well as the positions and orientation of the VR headset, controllers, and interactive objects. The positions of all the objects in the scene are updated by this component, and the pose of the headset and controller are used to animate a human avatar using the Final IK Unity package [23], to give the robot a realistic view of the human (see figs. 1 and 2). The final component is the ROS server, which uses ROS bridge [15] to connect the separate nodes, and is responsible for controlling the robot in the simulation.

### B. Data Collection

Participants were brought into a lab, and the simulation was explained by telling the user how the robot would prompt their response by asking for a description of the object. They were instructed to respond in a way that they would describe the object to someone that had never scene the object before.

People described objects in a variety of ways, giving verbal reports of of physical attributes, ideal usages, and origins of the object. There were fifteen participants in the study. They were between the age of 21 and 36 with a mean age of 25. Nine were male, six were female and one did not respond. Eight identified as Asian and seven as white.

After a familiarization period with the headset, the user was given two controllers. The right controller's trigger was to be held for the duration of an object's description, serving as a signal to start and stop a data collection instance. The data collected on head pose was recorded by Unity and transformed into the robot's frame of reference in Robot Operating System (ROS) Melodic. After hearing the instructions with an opportunity for questions, the users were given the virtual reality headset, and taught how to adjust the interpupillary distance of the screens, height of the headset while resting on the head, cranial length, and fold-down headphones.

The objects described by the user were captured as point cloud data from a simulated Xbox Kinect. The point clouds were segmented and clustered in order to detect where objects on the kitchen countertop were positioned. Following the final description, the users were thanked by the robot for completing the study and then filled out a post-study survey. Prior to exiting the lab, participants were informed that the study was not fully autonomous, as the robot had a conductor for a wizard of oz operation. Conductors were present with the participants for the duration for the experiment, and while the headset includes headphones, they do not cancel sound, so noise in the lab was audible.

### C. Gaze Calculation

During experiments the entire interaction was captured in ROS, including the audio from the user, simulated RGB and depth from the robot's perspective, point clouds, the position

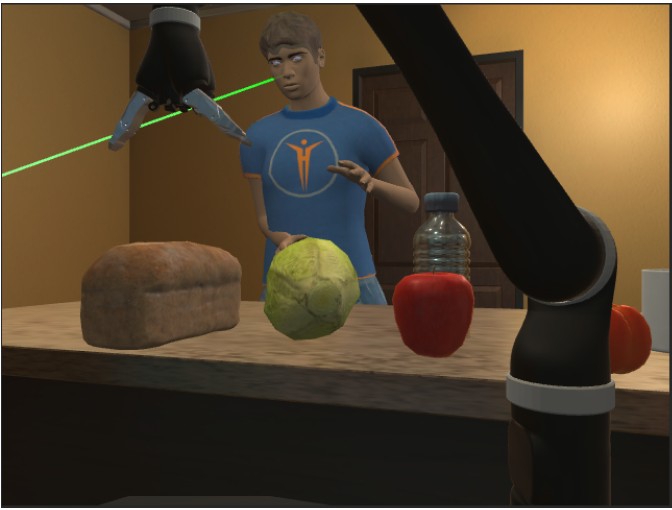

Fig. 1. The robot's perspective when prompting the user for a description of the bread. The gaze line (not visible to participants) is overlaid on the image as transformed from the virtual reality headset's tracking to demonstrate the gaze-to-object proximity visible from the robot's sensors.

and orientation of all the interactive objects in the scene, and all the ROS transform messages. The point clouds were segmented using PCL to get point clouds for each object on the table that the robot can see. The position and orientation of all interactive objects and headset are captured in the bagfile in the Unity coordinate system. The origin of the odometry frame of the robot in ROS matches the origin of the global coordinate system in Unity, but Unity uses the $z$ axis as forward/backward, $x$ as left/right and $y$ as up/down. Transforming the Unity positions and orientations to the robot's odometry frame of reference can be done by swapping axes and can be further transformed into the frame of reference of the point clouds captured by the robot using ROS transform libraries.

Once the head pose is in the same frame of reference as the point clouds, the position and orientation of the headset is used to determine the gaze direction by using a point one meter in front of the headset position in the direction the headset is facing. The vector defined by these two points is assumed to be the gaze vector. Each point in the point cloud is assigned a label by checking for the closest Unity object. The cosine similarity between the gaze vector and the vector between the head and the point is computed. Figure 4 shows this distance for each object on the countertop. The object containing the point with the smallest distance is the object that is assumed the person is looking at. Figure 3 shows the ground truth of the object the robot is indicating (blue bars) versus the object which the gaze vector intersects (other colors).

A ROS message is published that contains the raw audio that was recorded while the user is describing objects. In a few cases, the participants did not realize they should provide descriptions for the object the robot was indicating; in these cases the audio recording was used to manually annotate which object they were describing.

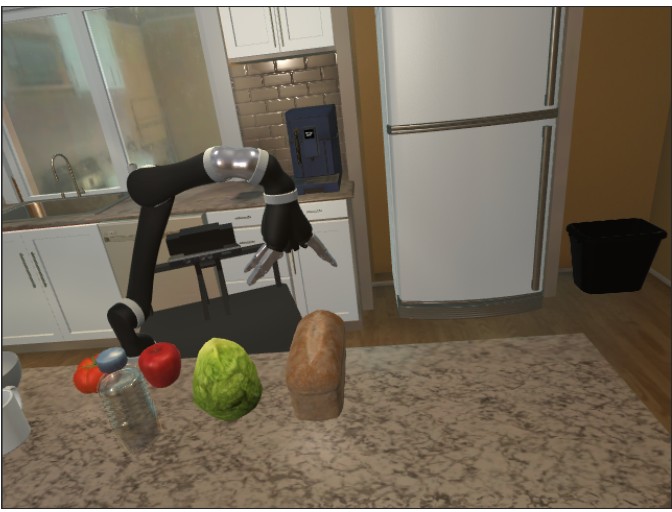

Fig. 2. The user's perspective of the robot moving its arm over the bread to ask the user for a description. The robot's audio informs the user of the robot's desire to know about the object and the movement of the arm indicates which object is of interest.

## IV. RESULTS

In order to determine how accurately our head pose-induced system identified the target object, we calculated the mean reciprocal rank of the distance calculation. Here we predict the rank of all objects based on their distance from the gaze vector, and then the inverse rank of the desired objects in all queries are averaged. For example, if the model predicts that the person is looking at the correct object according to ground truth, MRR $= \frac{1}{1} = 1$, a perfect score. This metric is suitable for capturing the intuition that some incorrect predictions are 'closer' than others. Table I shows how often the correct object was in the top 1, top 2, top 3, and the mean reciprocal rank (MRR) for all the participants. The mean percentage of time in which the object was correctly identified (top 1) was 51%, top 3 was 86%, and the mean MRR was 0.70.

A different perspective with a similar analysis format is shown in table II, where the ranks of interest are from the objects rather than the participant. From this it is clear that some objects, typically the larger or closer ones, were easier to correctly identify as gaze targets. This is intuitive given the distance metric we have chosen. For example, considering only the top prediction, lettuce was correctly identified in 98% of trials, while the mug—which was small and surrounded by larger objects—was correctly predicted only 2% of the time.

The closest object to the participants' gaze vector does tend to match the object they are describing. However, it matches better for larger objects and objects that are not clustered tightly together. One of the potentially confounding issues is the water bottle. Since the body of the bottle is transparent, there is a tendency to estimate that the user is looking through it to objects behind it. (Unity does not produce depth points for transparent objects, which is consistent with the real-world behavior of the depth sensors we intend to transfer learned models to; as a result, the point cloud of the water bottle contains few points and mostly consists of the cap.)

These results are consistent with previous physical gaze-tracking work, which show that looking at object separated by over $20°$ results in a viewer moving their head, and for objects

| Participant | Top 1 | Top 2 | Top 3 | MRR |
|---|---|---|---|---|
| 1 | 0.54 | 0.83 | 0.88 | 0.73 |
| 2 | 0.32 | 0.57 | 0.60 | 0.55 |
| 3 | 0.64 | 0.86 | 0.89 | 0.78 |
| 4 | 0.73 | 0.87 | 0.91 | 0.83 |
| 5 | 0.72 | 0.87 | 0.99 | 0.84 |
| 6 | 0.43 | 0.82 | 0.96 | 0.68 |
| 7 | 0.50 | 0.76 | 0.92 | 0.70 |
| 8 | 0.53 | 0.73 | 0.87 | 0.70 |
| 9 | 0.58 | 0.85 | 0.92 | 0.76 |
| 10 | 0.35 | 0.68 | 0.74 | 0.59 |
| 11 | 0.50 | 0.64 | 0.85 | 0.67 |
| 12 | 0.52 | 0.77 | 0.85 | 0.71 |
| 13 | 0.52 | 0.78 | 0.83 | 0.70 |
| 14 | 0.43 | 0.65 | 0.80 | 0.63 |
| 15 | 0.58 | 0.84 | 0.98 | 0.76 |
| Mean | 0.51 ± 0.12 | 0.76 ± 0.1 | 0.86 ± 0.1 | 0.70 ± 0.08 |

TABLE I
TOP 1, 2, 3 AND MRR RESULTS FOR ALL PARTICIPANTS

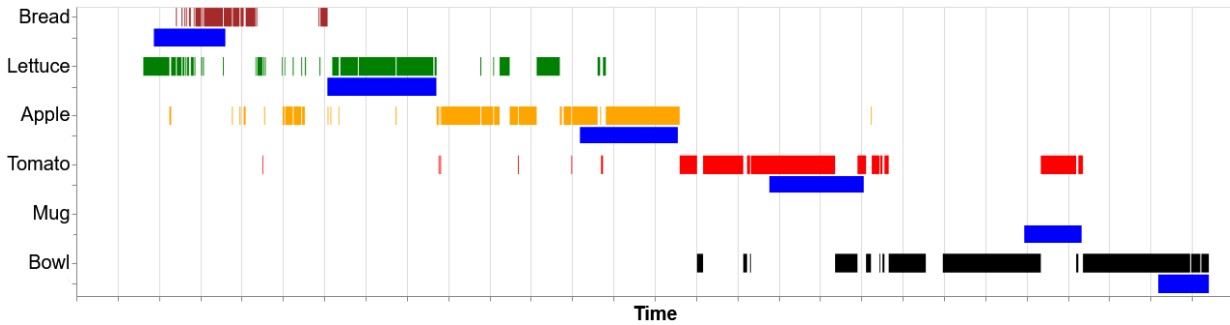

Fig. 3. For each object described by the user, the solid blue block indicates the ground truth of what is being described, starting at the timestamp when the user began their description of the object. Concurrent with the blue blocks are various colors, signifying the objects in which the user was looking as observed by head pose tracking. The breaks in the colors show how the user in this instance was looking from object to object even while describing a single object. The *y* axis is the objects that are being described, and the *x*-axis is time. This graph does not include 'distraction' objects not described by participants, which included a water bottle, drill, hammer, and first aid kit.

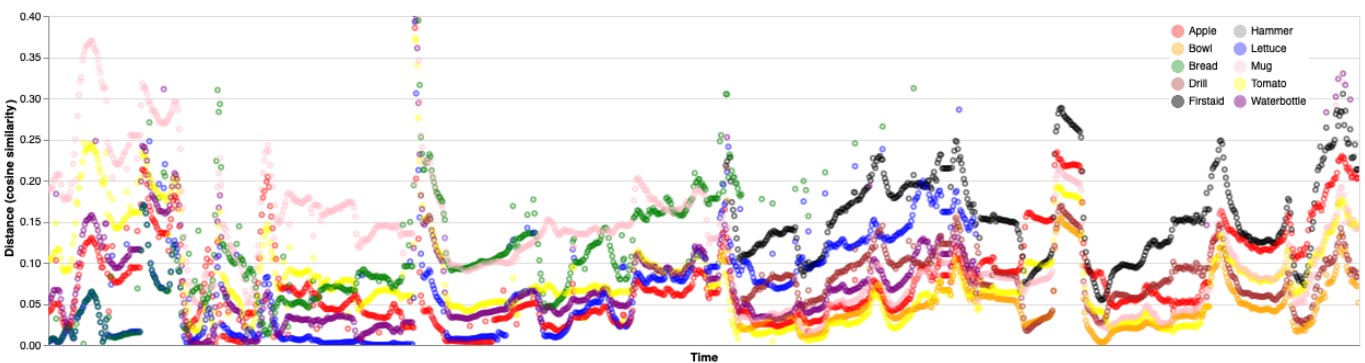

Fig. 4. The distance of the calculated gaze from each object over time, measured as cosine distance between the head pose vector and the vector between the participant's head and the objects. Different colors represent different objects; when the gaze coincides with an object, the distance drops to zero.

closer that $20°$ just moving their eyes while keeping their head steady [5]. This does demonstrate that the approach described in this work is faithful to the real-world gaze tracking we are attempting to simulate.

*A. Discussion*

The results of the study are promising in that they show the extraction of object labels in real time based on a head-pose as a proxy for gaze is feasible. Anecdotally, we discovered that participants looked at some objects more often than others, likely accounted for by varying object size and visibility; understanding this effect is one target of future work. One hypothesis is that certain objects require more visual or cognitive processing, e.g., based on the complexity of the object in

| Object | Top 1 | Top 2 | Top 3 | MRR |
|--------|-------|-------|-------|-----|
| Tomato | 0.77 | 0.88 | 0.94 | 0.84 |
| Lettuce | 0.98 | 0.99 | 0.99 | 0.99 |
| Apple | 0.30 | 0.77 | 0.98 | 0.60 |
| Mug | 0.002 | 0.06 | 0.33 | 0.12 |
| Bowl | 0.60 | 0.85 | 0.92 | 0.75 |
| Bread | 0.44 | 0.96 | 0.98 | 0.71 |
| Mean | $0.51 \pm 0.35$ | $0.75 \pm 0.35$ | $0.86 \pm 0.26$ | $0.67 \pm 0.3$ |

TABLE II
MRR RESULTS FOR ALL OBJECTS

shape, origin, or usage. Another possibility is the vibrancy of the colors in the objects that drive the vision to the object as a matter of immediate interest and attention captivation. Overall, while work remains, this approach to simulating gaze tracking in simulation shows promise as one of several modalities for human-robot interaction and particularly object selection.

## V. CONCLUSION AND FUTURE WORK

The results show promise in using head pose in VR as a proxy for gaze tracking when understanding what object is the focus of a person's attention. In future, we will integrate our approach with work on learning to understand language about objects based on descriptions and physical interactions with those objects. Future work also includes comparing the results of using head pose to determine gaze direction in virtual reality with results from the same experiment done in the real world with a physical robot, using identical head pose/gaze vector models.

A further refinement for future work is to consider the same concept of gaze as determined by head pose but use a hidden Markov model to determine what object is being targeted based on the gaze vector. This would allow for more sophisticated understanding of the probable focus of the gaze, and reduce artifacts such as favoring larger, closer objects as predicted targets.

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
