# OpenReview forum: "Head Pose as a Proxy for Gaze in Virtual Reality"
_humanrobotinteraction.org/HRI/2022/Workshop/VAM-HRI — VAM-HRI 2022_

### Official Review · Reviewer_Kzot · 2022-02-23
**Great Data Collection Idea / May Benefit from Some Robotics Framing**

**Rating:** 8
**Confidence:** 5

**Review:**

Summary:
This work discusses a data collection study (n=9) for measuring head pose as a proxy for eye gaze in a robot natural language grounding task. The study collected many types of data (well described within the paper), labelled times participants stared at/described an object, and measured cosine similarity of the desired object's position vector vs gaze vector (all relative to the headset). There were promising results around larger/closer/more salient objects with some larger difficulties for the opposite. Given a bigger early spin toward the HRI task in the title/intro (see feedback), I believe this would make a great contribution toward the VAM-HRI workshop.

Feedback:
I think this is a very cool idea and very interesting paper (wonder if it can also be expanded to phones or if the modality doesn't transfer...). The following feedback is aimed for improvements/possible future work/discussion at the workshop.

Is there a reason you couldn't also use a VR headset with eye gaze onboard as another measure of "ground truth" (e.g., Vive Pro Eye)? A small concern I have is around people not consistently looking directly at the object (e.g., when someone looks away to think). A big point of feedback for this work is better incorporating the VAM + robotics component of the work (at least into the title/introduction as the title + intro reads as a VR paper/not about grounding in robotics). I believe there is work on head pose as gaze estimation for many tasks especially in VR (https://scholar.google.com/scholar?hl=en&as_sdt=0%2C5&q=head+pose+as+gaze+estimation+vr&btnG=). Also it seems this is  for a specific task but that is only mentioned starting in section 2. Also if you haven't already, I would look into Eric Rosen's (et al.) work on gaze for object selection as it is very relevant (http://cs.brown.edu/people/er35/publications/pvd_iros2020.pdf). Very interesting discussion around the object's salience as well, would love to see more work around measuring that metric semi-objectively (e.g., how similar objects are/small objects = more difficult labelling of eye gaze).
More fine grained feedback below.
- citation for "Gaze tracking, while valuable, generally requires specialized hardware and often negatively affected by external features such as eye color"
- what experience did participants have with VR?
- table II should possibly be pushed to the top/somewhere before references

---

### Official Review · Reviewer_uKPE · 2022-02-25
**Interesting and useful study, clear accept**

**Rating:** 9
**Confidence:** 5

**Review:**

The proposed study for evaluating how effective head pose is as a proxy for eye-gaze in VR object selection tasks is interesting and relevant to the VAM-HRI community. The paper is also well-written and clear, so this is a clear accept.

Feedback:
1. The introduction mentions that there are connections to “building an unsupervised learning pipeline for object descriptions”, could this be clarified more? If the user is labeling objects with their head pose (as a proxy for gaze), how is this unsupervised?
2. The authors should cite [1], [2] because they are very relevant related works. Both of these works propose methods for tracking eye-gaze with a MR headset and inferring what object a user is intending to look at by using (or building on top of) hidden markov models. They do not use head-pose as a proxy for eye-gaze, but they are extremely relevant.
3. The sentence in the related work should be “the accuracy did not differ when gazing at…”, not as.
4. The MRR acronym should be defined first before being mentioned in Section 4.

[1] Puljiz, David, et al. "HAIR: Head-mounted AR Intention Recognition." arXiv preprint arXiv:2102.11162 (2021).
[2] Rosen, Eric, et al. "Mixed reality as a bidirectional communication interface for human-robot interaction." 2020 IEEE/RSJ International Conference on Intelligent Robots and Systems (IROS). IEEE, 2020.

---

### Decision · Program_Chairs · 2022-03-04

Accept